



# New insights on particle characteristics of previously characterised EGRIP ice core samples via single particle ICP-TOFMS

Nicolas Stoll[1,2], David Clases[3], Raquel Gonzalez de Vega[3], Matthias Elinkmann[3], Piers Larkman[1,4], and Pascal Bohleber[1,4,5]

[1]Department of Environmental Sciences, Informatics and Statistics, Ca'Foscari University of Venice, Venice, Italy
[2]Department of Earth and Space Sciences, University of Washington, Seattle, USA
[3]Nano Micro LAB, Institute of Chemistry, University of Graz, Graz, Austria
[4]Department of Geosciences, Alfred Wegener Institute Helmholtz Centre for Polar and Marine Research, Bremerhaven, Germany
[5]Department of Geosciences, Goethe University Frankfurt am Main, Germany

**Correspondence:** Nicolas Stoll (nicolasangelo.stoll@unive.it)

**Abstract.** Polar ice cores contain an archive of chemical impurities, which can be used as a proxy for the past climate. State-of-the-art chemical methods increase our knowledge about these impurities, especially when applying a cascade of complementary techniques to the same samples. Single particle inductively coupled plasma-time of flight mass spectrometry (SP ICP-TOFMS) has yet to be fully utilised to study polar ice; only two studies have described its application so far. This is surprising given

its capability to access critical physicochemical parameters of insoluble particles, such as number concentration, the mass and size distributions and elemental composition of deposited particles. In this study, we demonstrate the analysis of ice core samples, which have previously been characterised with Raman spectroscopy and laser-ablation inductively coupled plasma mass spectrometry, via SP ICP-TOFMS. By investigating melted samples, new possibilities for the in-depth geochemical analysis of deposited aerosols arise. We analyse nine samples from the EGRIP ice core from different climate periods throughout the

last 50 ka. Samples from cold periods, such as Glacial Stadials, generally have the highest particle concentrations, especially when containing cloudy bands. We present an overview of the different particulate elements, which are largely unexplored in deep polar ice cores. We introduce a novel approach to estimating particle size by transferring mineralogy information from Raman spectroscopy, leading to a more accurate particle size representation via SP ICP-TOFMS, especially in the nanometre range. Our results show the largely untapped potential of SP ICP-TOFMS and demonstrate new opportunities to analyse intact

particles as proxies for the palaeoclimate. The combination of impurity characterisation methods applied to the same samples empowered us to gain complimentary perspectives on particles and collect a comprehensive data set from the same samples. In view of the ongoing endeavour to retrieve the "Oldest Ice", SP ICP-TOFMS may become a critical tool to access vital information and new depths of insights.

## 1 Introduction

Chemical impurities in polar ice cores deliver essential information about the climate of the past, dust sources, and atmospheric transport paths (e.g., Legrand and Mayewski, 1997). They are further influential components within the ice microstructure im-



pacting, among others, the deformation of ice (e.g., Paterson, 1991; Stoll et al., 2021b). Different techniques are applied to measure ice core samples depending on the focus of the investigation. Bulk impurity concentrations are usually measured with continuous flow analysis (CFA) via melting and measuring the uncontaminated inner parts of an ice sample (e.g., Röthlisberger

et al., 2000; Kaufmann et al., 2008). Non- or micro-destructive methods on discrete samples are, for example, cryo-Raman spectroscopy (e.g., Fukazawa et al., 1997; Ohno et al., 2005; Stoll et al., 2022), scanning electron microscopy (SEM) (e.g., Barnes et al., 2002), and laser-ablation inductively coupled plasma mass spectrometry (LA-ICP-MS), which can measure the distribution of impurities in one or two dimensions (e.g., Della Lunga et al., 2017; Bohleber et al., 2023). Coupling these methods in series allows complimentary data collection across the same samples. It provides opportunities to gain a better understanding

of the underlying geochemistry and the localisation of impurities in polar ice (e.g., Stoll et al., 2023b). Instrumentation and methodology are rapidly advancing and new measurement-based modelling approaches are on the way (Larkman et al., 2024) to provide new insights into the spatial distribution of elemental impurities. However, due to the particular material characteristics of ice, the relatively small research field of ice core science, and the needed analytical expertise, it can take years or even decades to implement state-of-the-art chemistry techniques into routine settings for ice core research.

Elemental mass spectrometry and specifically inductively coupled plasma-mass spectrometry (ICP-MS), is an established analytical technique in many research fields (e.g., Clases and Gonzalez de Vega, 2022a, b), providing isotope and elemental information over a large dynamic range. However, over the last two decades, the so called "single particle (SP) mode" has been gaining momentum (e.g., Bolea et al., 2021; Laborda et al., 2023). In this mode, mainly inorganic particles, ranging from nano- to microparticles, are introduced individually into the plasma and are atomised and ionised to form secluded clouds of

elemental cations. Following extraction and by using rapid mass analysers and detectors, each ion cloud can be detected with several data points, and spatially resolved "particle events" are registered at the detector. Here, the frequency and intensity of events can be calibrated into particle number concentrations and the elemental mass per particle, respectively, if suitable standards are analysed concurrently. If the mineral phase of the particle is known (and therefore elemental mass fractions and phase density), size distributions may be modelled. The dissolved forms of an analysed element are derived from the

background signal (baseline) due to its homogeneous distribution in the aerosol droplets. Due to the high counting rates and the ability to determine natural abundances of particles in complex environmental matrices, this technique is increasingly utilised in environmental sciences. Despite its potential to identify large numbers of solid inclusions, such as dust particles, ice core scientists have hardly exploited SP ICP-MS.

Over the last years, time of flight (TOF)-based analyses provided a paradigm shift for the analysis of single particles.

Unlike its scanning counterparts (quadrupole) it provides a (quasi-)simultaneous detection mechanisms enabling the analysis of virtually all elements of the periodic table contained in single particles. For ice core research, ICP-TOFMS is arguably the ideal technique for the fast mapping of the microstructural distribution of chemical elements when combined with LA (Bohleber et al., 2021; Stoll et al., 2023a) or for the counting and characterisation of particulates when operated in SP mode. Although quadrupole-based methods are generally more sensitive for individual isotopes, SP ICP-TOFMS provides both post-

processing options and specific hardware features for a competitive trace element analysis (Lockwood et al., 2024b). However,



more importantly it enables the unique option to carry out non-target particle screening (Gonzalez de Vega et al., 2023) and to determine the internal and external mixing state of particles (Tharaud et al., 2022).

SP ICP-TOFMS is increasingly used in a variety of disciplines, ranging from medical to material science (see examples in e.g., Clases and Gonzalez de Vega, 2022b). However, to our knowledge, SP ICP-TOFMS has, so far, only been applied to ice cores in two peer-reviewed studies. Modern-era ice from the Disko Island ice cap (DS14) and an inland site on the Greenland ice sheet (GW14) were analysed by Osman et al. (2017). The second study (Erhardt et al., 2019) mainly described the extension of the Bern CFA system by investigating a continuous section of late Holocene ice from the East Greenland Ice-core Project (EGRIP) ice core, the first deep ice core drilled through an ice stream (Stoll et al., 2024). Both studies focused primarily on the analytical challenges and advantages of the technique and a few major elements (e.g., Fe, Mg, Al) or particle classes for source region identification. Even though additional work on the Dome C ice core is on the way (Lee et al., 2024) data remain sparse.

Recent studies on the EGRIP ice core, utilising different optical and chemical methods, such as confocal cryo Raman spectroscopy and LA-ICP-MS 2D Imaging, shine new light on the microstructural localisation of insoluble and soluble impurities by systematically analysing samples from different climate periods (Stoll et al., 2021a, 2022, 2023b; Bohleber et al., 2023). These efforts lead to progress in the generalisation of localisation trends (Stoll et al., 2023a) and the development of homogenous ice standards enabling the intensity output to be calibrated to concentrations and thus, a better comparison between future CFA and LA-ICP-MS measurements (Bohleber et al., 2024). An essential asset of Raman spectroscopy and LA-ICP-MS is that they are non- or micro-destructive, enabling further use of the analysed samples. Here, we apply SP ICP-TOFMS to nine previously characterised samples to the best degree possible to better understand and highlight the contribution of this state-of-the-art method to investigating insoluble particles as the final step in a cascade of different analyses. Our objective is to extend the limited available data regarding the characterisation of insoluble particles in polar ice focusing on the most abundant elements. Building on the characterisation of certain minerals via Raman spectroscopy in previous studies on the same samples (Stoll et al., 2022, 2023b), we further introduce an approach to estimate particle sizes from SP data. Reliable dust particle size data are crucial, for instance, for modelling global climate change due to the radiative effect of mineral aerosols in the atmosphere and their impact on other aspects of the climate system, such as biogeochemistry and cloud nucleation (e.g., Tegen and Lacis, 1996; Mahowald et al., 2014; Adebiyi and Kok, 2020). Furthermore, dust might play a vital role in reconstructing the age of million-year-old ice by comparing the ice core dust flux with dust proxies from marine sediment cores (Martínez-Garcia et al., 2011; Wolff et al., 2022). The here presented multi-method and maximum-output strategy is thus a valuable approach for analysing precious ice samples, such as from the ongoing "Oldest Ice" quest (Fischer et al., 2013).

## 2 Methods

### 2.1 Samples

We used nine samples from the EGRIP ice core, which we previously analysed regarding their microstructure (microstructure mapping), and insoluble, micrometre-sized (confocal cryo-Raman spectroscopy) and general (LA-ICP-MS 2D imaging) impurity localisation (Stoll et al., 2021a, 2022; Bohleber et al., 2023; Stoll et al., 2023b). The shallowest three samples were



labelled as S1, S7, and S10 in Stoll et al. (2022), and deeper ones were S1, S2, S7, S8, S11, and S13 in Stoll et al. (2023b).
We re-labelled them according to their climate period plus a chronological number increasing with depth/age. The youngest
sample is from the Holocene and thus H1, while the deepest is from the last glacial and thus G9 (Table 1). Details on depth,
age, and minerals of interest as previously identified via Raman spectroscopy are also displayed in Table 1. Insoluble particle
concentration data are only available for the shallowest three samples (H1, H2, YD3) (Stoll et al., 2021a), which showed very
low concentrations except for YD3. This sample originates from the Younger Dryas (GS-1), thus having a very high insoluble
particle concentration, further explored in a detailed LA-ICP-MS investigation focusing on cluster of Fe and Al interpreted as
insoluble particles (Bohleber et al., 2023). For samples G4-G9, only visual stratigraphy data are available as insoluble particle
proxy (Stoll et al., 2023b). These data show no features (G4), weak cloudy bands (G5 and G9), i.e. layers of high insoluble
particle content, intermediate cloudy bands (G6 and G7), and strongly defined cloudy bands (YD3 and G8).

**Table 1.** Analysed EGRIP samples and respective properties. Further information on the first three samples can be found in Stoll et al. (2022)
and the remaining six in Stoll et al. (2023b). YD3 was investigated in detail by Bohleber et al. (2023).

| Sample | Depth (m) | Age b2k (ka) | Climate period[+] | Cloudy band | Identified relevant minerals[*] |
|---|---|---|---|---|---|
| H1 | 138.9 | 1.0 | Holocene | - | quartz, feldspar, mica |
| H2 | 900.0 | 7.6 | Holocene | - | quartz, feldspar, mica |
| YD3 | 1256.9 | 12.1 | GS-1 (YD) | + | quartz, feldspar, mica, hematite, anatase |
| G4 | 1360.8 | 14.4 | GI-1e | - | quartz, feldspar, mica, hematite, anatase, rutile |
| G5 | 1367.1 | 14.5 | GI-1e | + | quartz, feldspar, mica, hematite, rutile |
| G6 | 1823.5 | 34.7 | GS-7/GI-7a | + | quartz, feldspar, mica, hematite, rutile |
| G7 | 1883.1 | 37.3 | GI-8c | + | quartz, feldspar, mica, hematite |
| G8 | 2016.0 | 44.0 | GS-12 | + | quartz, feldspar, mica, hematite, magnetite |
| G9 | 2115.1 | 49.8 | GI-14a | + | quartz, feldspar, mica, hematite |

b2k: before 2000 CE after Mojtabavi et al. (2020) and Gerber et al. (2021), GS: Glacial Stadial, GI: Glacial Interstadial, YD: Younger Dryas, [+]: after Rasmussen
et al. (2013), [*]: identified by Stoll et al. (2022) and Stoll et al. (2023b) and chosen for particle size estimates; does not represent all characterised minerals in these
samples.

## 2.2 Single particle ICP-TOFMS measurements

Analysis was carried out over three sessions at the Nano Micro LAB at the Institute of Chemistry, University of Graz, Austria.
All samples were transported from Venice, Italy, to Graz, Austria, in cooled polystyrene boxes, and no melting indications
were observed. Solid ice samples were rinsed several times with Milli-Q ultrapure water (MQ) in a laminar flow cabinet for
decontamination and melted in vials at room temperature. The workflow is schematically displayed in Fig. 1.

A Vitesse ICP-TOFMS system by Nu Instruments (Wrexham, UK) was operated in SP mode and recorded and binned mass
spectra from 20-240 amu, every 100 $\mu$s (4 spectra binning) before saving data to disc while blanking the range 31.5–38.5 amu
due to a high signal from the oxygen dimer to avoid signal saturation at the detector. Each sample, blank (frozen MQ water let to





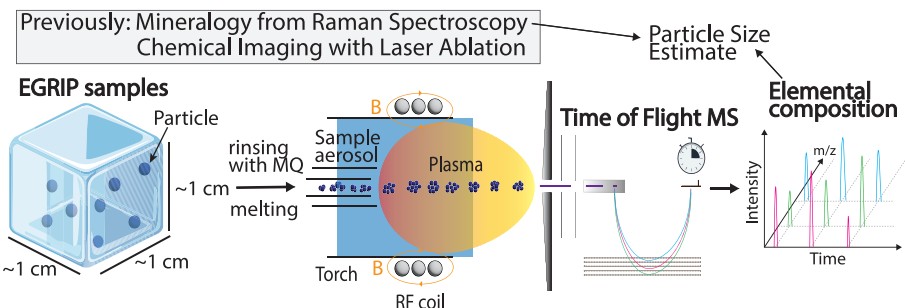

**Figure 1.** Schematic overview of the workflow and set-up of SP ICP-TOFMS. Ice core samples are decontaminated with MQ water and melted. The sample aerosol is introduced via liquid nebulisation into the ICP for ionisation. Time of Flight measurements resolve the elemental composition of particles (not to scale) previously stored in the ice. Mineralogy data from Raman spectroscopy enable the estimate of particle sizes. m/z: mass-to-charge ratio, B: magnetic field, RF: Radio Frequency, MS: Mass Spectrometer.

melt), and calibration standards were recorded for 100 s. The ICP-TOFMS instrument was equipped with a concentric nebuliser (Glass Expansions) and a cooled (5°C) cyclonic spray chamber. The plasma was operated at 1.35 kW, and the segmented reaction cell was operated with He and $H_2$ flow rates of 12 and 7 mL/min, respectively. The typical nebuliser flow rate was

approximately 1.16 L/min. Data acquisition was performed using the Nu Codaq software (Nu Instruments). Transport efficiency was 5.6% and was determined by analysing Au Nanoparticles (nanoComposix, US) and ionic standards with a known mass and concentration, respectively, using an automated approach via the open-source SP data processing platform SPCal (Lockwood et al., 2021, 2024a). Usually, Compound Poisson statistics were used to distinguish ionic background and noise from SP events with an $\alpha$ value of $1e^{-6}$. We applied automatic thresholds for all data except particle size. In cases with higher background

(Lockwood et al.), Gaussian statistics were more adequate and selected, respectively. To enable a comparison across samples with different ionic backgrounds and therefore, with different size detection limits, we used a conservative approach, in which the highest background across all samples was used to define a common threshold of all remaining samples (Table A1 and A2). While this prevents the detection of small particles and therefore underestimates number concentrations and overestimates size distributions, it allows to compare the occurrence and distribution of particles in a common size window.

## 2.3 Single particle data processing and analysis

To process the data, we applied SPCal version 1.2.7, designed to process and analyse SP data (Lockwood et al., 2021, 2024a). Estimating particle sizes is possible if specific crystal phases are chosen for each element to obtain phase density and element mass fractions (Lockwood et al., 2024b). The measured MQ was used to obtain a blank value, and determined ionic background and particle numbers were subtracted for each analysed element. Gold standards were interrogated frequently to monitor

transport efficiency. We focused on the most abundant elements $^{24}$Mg, $^{27}$Al, $^{28}$Si, $^{48}$Ti, and $^{56}$Fe. For the analysis of $^{48}$Ti, $^{48}$Ca may potentially interfere. However, we have not detected any significant impact, which was evaluated by monitoring the





$^{47}$ Ti/$^{48}$ Ti ratio. We also show less common elements, such as $^{90}$ Zr or $^{139}$ La, depending on their occurrence in the respective samples.

## 2.4 Particle signal losses and overlap

We used a Poisson-based model to estimate the loss of particle signals due to event overlap as suggested by Peyneau and Guillon (2021). We chose a sample with a low event rate (H1) to estimate the mean peak width at base for each element for all other samples. This approach was chosen as the best estimator for the peak width, as higher event count rates introduce a bias that overestimates peak widths (Peyneau, 2022). Importantly, these studies showed that the distribution of the event durations does not matter as only the expected values (i.e., the average) are needed, making the approach very stable.

In the data, depending on the element, we observed a mean peak width at base from 206 $\mu$s ($^{24}$ Mg) to 632 $\mu$s ($^{56}$ Fe), which agrees reasonably well with peak width values reported in the literature (e.g., Fuchs et al., 2018). Next, "true" particle event rates up to 4000 $s^{-1}$ with spacing of 0.01 $s^{-1}$ were calculated. For each entry, the observable particle count rate was calculated, considering losses from event overlap (Peyneau and Guillon, 2021) (Table 2). The highest tolerated count rate, i.e., the upper particle number concentration threshold, was defined in the following way: When the quotient of the observed count

rate divided by the true count rate was lower than 90%, this observed count rate was defined as the upper number concentration threshold. For samples with more particles than the respective threshold, the data can be assumed to be biased towards 1) fewer particles, 2) apparently larger particles, and 3) apparently more mixed composition particles. We marked samples and particle species reaching this threshold and it should be emphasised that reported value are presenting a conservative measure underestimating number concentrations and overestimating sizes.

**Table 2.** Thresholds for the maximum observable event rate (NP/s) and the maximum observable absolute particle count per sample for the most abundant elements.

| Species | Maximum event rate (NP/s) at 90% deviation | Maximum particle count per sample at 90% threshold (100 s) |
|---|---|---|
| $^{24}$ Mg | 294.9 | 29490 |
| $^{27}$ Al | 188.9 | 18887 |
| $^{28}$ Si | 182.6 | 18261 |
| $^{48}$ Ti | 132.1 | 13207 |
| $^{56}$ Fe | 126.9 | 12687 |

## 145 2.5 Particle size estimate

The estimation of particle sizes from SP ICP-TOFMS data requires knowledge about the particle species, i.e., the mineral to consider elemental mass fractions and density. For the most common elements $^{27}$ Al, $^{24}$ Mg, $^{28}$ Si, $^{48}$ Ti, and $^{56}$ Fe we assumed minerals based on previously conducted Raman spectroscopy analyses of micrometre-sized inclusions inside our



EGRIP samples identifying these minerals (Stoll et al., 2022, 2023b). These assumptions might differ on the nano-scale, but
the derived mineralogy from micro-inclusions via Raman spectroscopy is so far the best proxy. Stoll et al. (2022, 2023b)
identified quartz and members of the feldspar and mica group in all nine samples (Table 1). $SiO_2$ (quartz) was thus chosen for
[28] Si. For simplicity, the very abundant potassium feldspar ($KAlSi_3O_8$) and mica phlogopite ($KMg_3AlSi_3O_10(F)_2$) were
chosen for [27] Al and [24] Mg, respectively. Hematite ($Fe_2O_3$) occurred in all samples except the shallowest two and, thus, was
selected for [56] Fe. We chose $TiO_2$ (rutile) for [48] Ti, because it was the more abundant form of the two observed titanium-oxide
forms (rutile and anatase), which were identified in 13 samples analysed by Stoll et al. (2022, 2023b).

The chosen element-mineralogy relationship is a simplification and thus limited; it works as a first proof of concept. This
new approach integrates information from previous measurements and, therefore, enables an independent estimate of particle
sizes. SP ICP-TOFMS is useful to spotlight particles with dimensions at the nanoscale (a few nanometres to above 3 $\mu$m), a
size scale often overlooked and mostly uncharted in ice cores. Measurements cover more of the fine particle size range and
provide missing and more comprehensive information to established methods, such as Coulter Counter (usually 0.6-10 $\mu$m)
and laser particle detectors (usually 0.9-15 $\mu$m) (Vallelonga and Svensson, 2014).

### 2.6    Methodological challenges of SP ICP-TOFMS

The analysis of particles in natural systems is a challenging endeavour. It requires certain assumptions to establish suitable
bottom-up models on various particulate parameters, including size and mass distributions, particle number concentrations,
and internal and external mixing states. Determining the mass and size of single particles with SP ICP-TOFMS requires
knowledge of the intrinsic mineral phase. Alone, SP ICP-TOFMS may only determine the mass of elements within a particle.
Any information on particle species, size, and shape is lost upon the atomization step in ICP. However, it is possible to obtain
critical species-specific parameters beforehand by applying complementary techniques. Cryo Raman spectroscopy can carry
out in-situ mineral characterization, which helps link detected elements with specific species and crystal phases and estimate
the elemental mass fractions. Especially in scenarios in which multiple polymorphic phases are present (e.g., rutile and anatase
($TiO_2$)) or in which only a small number of particles is counted, compromises in the modelling of size distributions are
required. Also, we emphasize that current cryo Raman systems and SP ICP-TOFMS focus on different size scales, which is
interesting for complementary analysis but limiting when the same particle fractions shall be characterized in a multi-modal
fashion. Further, SP ICP-TOFMS only investigates melted ice, meaning that soluble particle fractions or agglomerates may not
be conserved during the melting process.

## 3    Results

### 3.1    Particulate concentration and normalised detections

Absolute SP detections depend on the chosen ionic background threshold, which is either 1) automatically chosen (Sect. 2.2)
or 2) can be set manually. As different thresholds for the ionic background of each element are calculated in each sample,





comparing absolute detections quantitively among samples is tricky. However, a qualitative analysis can be conducted by utilising normalised detections. Here, independent of the total detections, the relative shares of each element per sample deliver a basis for the comparison of our samples. We discuss the challenges of choosing the fitting threshold approach in Sect. 4.6.

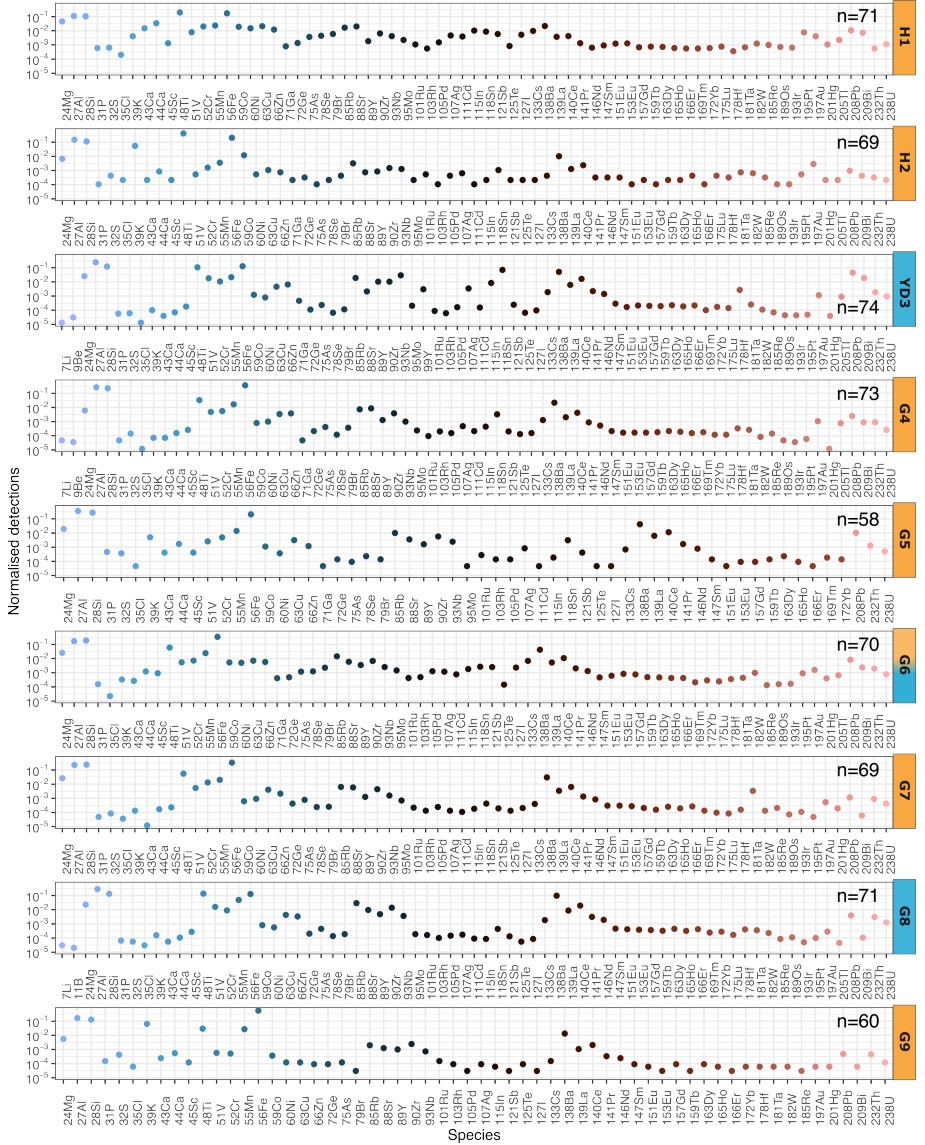

**Figure 2.** Normalised detections for all chemical species detected with the automatic ionic background threshold; n represents the number of particulate elements in the sample. Each sample is labelled with its sample ID from Table 1, and is filled with orange or blue to Holocene/Interstadial and Stadial samples, respectively. The debatable climate period of G6 is discussed in the text. Note the respective chemical species on the x-axis indicated by the same colours.





Up to 74 different particulate elements were present in the nine analysed samples. The normalised detections for all detected particulate elements are displayed in Fig. 2 and specific elements are discussed in more detail below. YD3 contains the largest number of particulate elements (n=74). G4 contains 73 particulate elements closely followed by H1 and G8 (both n=71) and G6 (n=70). The lowest numbers are found in G9 and G5, with 60 and 58 different elements, respectively. The least abundant elements (in normalised share) are $^{35}$ Cl (G4), $^{201}$ Hg (G4), $^{43}$ Ca (G7), and $^{7}$ Li (YD3).

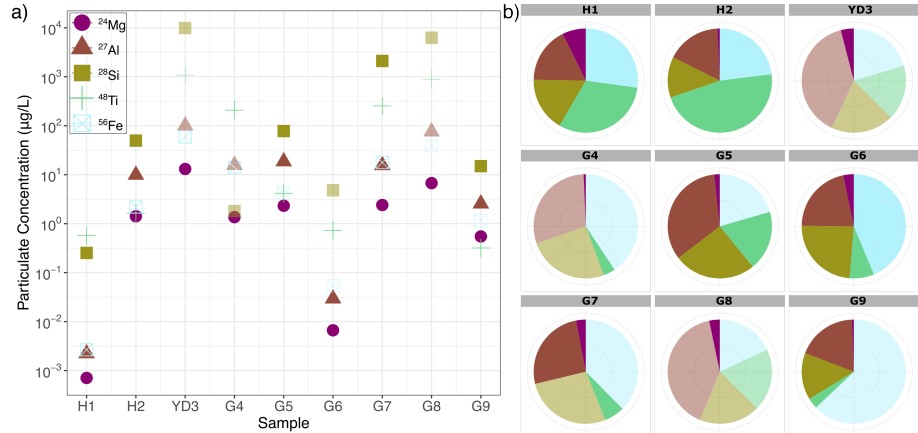

**Figure 3.** Derived single particle a) concentrations for $^{24}$ Mg, $^{27}$ Al, $^{28}$ Si, $^{48}$ Ti, and $^{56}$ Fe and b) normalised detections for each respective species and sample. Elements with particle detections above the calculated thresholds (Table 2) are indicated by lower opacity and mentioned in the text. The colour legend applies to both panels.

In the following we focus on the most abundant particulate elements $^{24}$ Mg, $^{27}$ Al, $^{28}$ Si, $^{48}$ Ti, and $^{56}$ Fe. The highest particulate concentrations were found in YD3 ($^{28}$ Si: $9.876 \times 10^3 \mu$g/L, $^{48}$ Ti: $1.076 \times 10^3 \mu$g/L) and G8 ($^{28}$ Si: $6.266 \times 10^3 \mu$g/L, $^{48}$ Ti: $8.851 \times 10^3 \mu$g/L) (Fig. 3a). The concentrations in H1 are by far the lowest ($^{24}$ Mg: $7.073 \times 10^{-4} \mu$g/L, $^{27}$ Al: $2.235 \times 10^{-3} \mu$g/L, $^{56}$ Fe: $2.668 \times 10^{-3} \mu$g/L) followed by G6 ($^{24}$ Mg: $6.644 \times 10^{-3} \mu$g/L, $^{27}$ Al: $2.940 \times 10^{-3} \mu$g/L, $^{56}$ Fe: $5.407 \times 10^{-2} \mu$g/L). Usually, $^{28}$ Si and $^{48}$ Ti show the highest concentration followed in decreasing order by $^{27}$ Al, $^{56}$ Fe and $^{24}$ Mg. The normalised data (Fig. 3b) show that $^{48}$ Ti is the dominant species, by particulate concentration, in the Holocene (H1, H2), $^{27}$ Al in the Younger Dryas (YD3), and $^{56}$ Fe in the last glacial (G4, G6, G7, G9) with the exception of G5 and G8 where $^{27}$ Al is more abundant. In G9, $^{56}$ Fe represents almost two-thirds of the total normalised particulate concentration share. $^{28}$ Si has the most constant share throughout all samples, and its abundance fluctuates roughly between one-third and one-quarter. $^{24}$ Mg is generally rare and hardly found in H2, G4, and G9. Detections above the event rate thresholds described in Sect. 2.4 were found in YD3 ($^{27}$ Al, $^{28}$ Si, $^{48}$ Ti, $^{56}$ Fe), G4 ($^{27}$ Al, $^{28}$ Si, $^{56}$ Fe), G5 ($^{56}$ Fe), G7 ($^{28}$ Si, $^{56}$ Fe), G8 ($^{27}$ Al, $^{28}$ Si, $^{48}$ Ti, $^{56}$ Fe), and G9 ($^{56}$ Fe).

To provide a rough source region proxy, we calculate the Fe/Al mass ratio (Table 3) (e.g., Jeong, 2008; Formenti et al., 2011; Erhardt et al., 2019). To further compare our samples with available data, we calculate the Fe particulate and ionic background. For simplicity, we used all particles containing Fe.





**Table 3.** Particulate and ionic background $^{56}$Fe signals and the Fe/Al ratio.

| Sample | $^{56}$Fe particulate (%) | $^{56}$Fe ionic background (%) | Fe/Al |
|---|---|---|---|
| H1 | 47.8 | 52.2 | 1.19 |
| H2 | 61.3 | 38.7 | 0.22 |
| YD3 | 52.2 | 47.8 | 0.59 |
| G4 | 77.7 | 22.3 | 0.89 |
| G5 | 32.7 | 67.3 | 0.23 |
| G6 | 75.1 | 24.9 | 1.84 |
| G7 | 79.2 | 20.8 | 1.14 |
| G8 | 35.3 | 64.7 | 0.52 |
| G9 | 87.2 | 12.8 | 0.45 |
| Mean | 61.0 | 39.0 | 0.79 |

## 3.2 Particle size estimate

### 3.2.1 Mean mass

To estimate particle sizes, density and element mass fractions must be considered (Sect. 2.3). For the five particulate elements
205 we focus on, we applied the highest thresholds as explained above. Mean masses range from 0.007 fg ($^{24}$Mg in H1) to 3078
fg ($^{28}$Si in YD3) (Fig. 4a). Particles containing $^{28}$Si and $^{48}$Ti are heaviest while $^{24}$Mg and $^{56}$Fe particles are the lightest. The
mean mass differs more between samples than between chemical elements inside the respective sample.

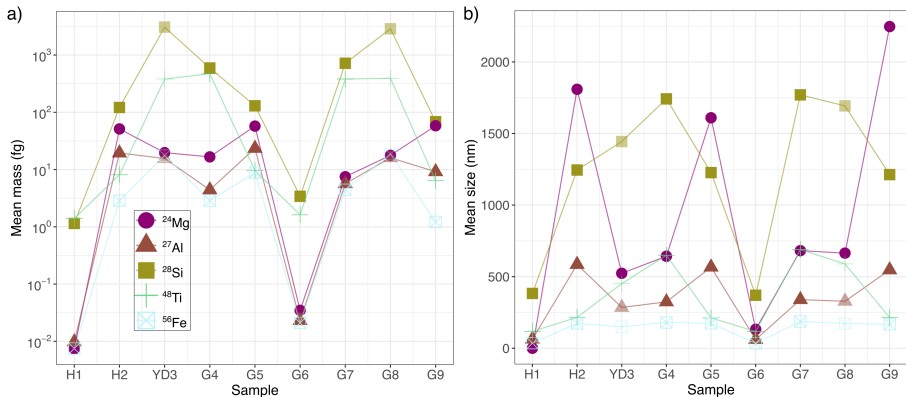

**Figure 4.** Single particle parameters calculated with uniform ionic background thresholds for $^{24}$Mg, $^{27}$Al, $^{28}$Si, $^{48}$Ti and $^{56}$Fe. a) Mean
mass measurements and b) mean size estimates. Size estimates are based on the chosen mineralogy as explained in Sect. 2.5. Opacity and
colour legend as in Fig. 3.





### 3.2.2 Mean and median particle sizes

Uniform ionic background thresholds for each element were applied to estimate particle sizes whilst considering mineral phases
previously identified with Raman spectroscopy (Sect. 2.2). Across all samples, we selected the highest ionic background to
make the detection of (large) particle comparable. Detections above the event rate thresholds (Sect. 2.4) were only reached in
YD3 ([27] Al, [28] Si, [48] Ti, [56] Fe) and G8 ([27] Al, [28] Si, [48] Ti, [56] Fe). While this approach enables the comparison of particle size
and number in a set window, it prevents the analysis of small particles, which could theoretically be detected using $\alpha=10^{-6}$.
Therefore, the following comparisons are merely qualitative.

The estimated mean and median particle sizes (Fig. 4b) and A1, respectively) show a wide range within 210 samples and over
the sampled spectrum and in the following comparisons and discussions, we will refer to mean values. Particles are comparably
small in samples H1 (36-383 nm) and G6 (34-370 nm). [28] Si-containing particles are the largest in six samples (H1, YD3, G4,
G6, G7, G8) and range from 370 nm (G6) to 1770 nm (G7). [24] Mg-particles are the largest in the remaining three samples (H2,
G5, G9) and range from 132 nm (G6) to 2247 nm (G9). [56] Fe-bearing particles are always the smallest, ranging from 34 nm
(G6) mm to 187 nm (G7). Median sizes are similar to the mean values (Fig. A1). In H1, the [24] Mg ionic background threshold
is too high and no particles were detected; particle sizes are thus plotted at 0 nm.

### 3.2.3 Particle size distribution

The samples, analysed with the highest ionic backgrounds, can be separated into two groups regarding particle size: small
(H1, G6) and large (H2-G5, G7-G9) (Fig. 4b). We show the size distributions of H1, G5, and G7 in Fig. 5. In H1 (Fig. 5a),
[24] Mg data are not available as the uniform ionic background threshold is too high. The LOD cuts off the finest spectrum for
[27] Al, which is otherwise lognormal distributed. [28] Si has very few detections with some outliers in the biggest size range.
[48] Ti, displays a lognormal distribution. [56] Fe also has a relatively small number of detections with slightly varying detections
roughly representing a lognormal distribution. In G5 (Fig. 5b), [24] Mg detections are low, within the micrometre range (1.4-1.9
$\mu$m) and with some outliers towards the largest sizes. [27] Al, [48] Ti, and [56] Fe are lognormally distributed. [28] Si has some outliers
of small size around its mean size. In G7 (Fig. 5c), [28] Si and [48] Ti particles are lognormal distributed and cover an extensive size
range reaching into the micrometre range. [27] Al and [56] Fe show lognormal distributions. [24] Mg has a lognormal distribution
with a few outliers towards larger sizes.

## 4 Discussion

### 4.1 Characterisation of particles and their link to climate

The concentration of [28] Si-bearing particles is highest in most samples (H2, YD3, G5, G7, G8, G9) (Fig. 3a). Given the
high abundance of silicate minerals, such as quartz (90% in the Earth's crust), this was an expected result. Crustal material
has been transported to Greenland regularly (e.g., Steffensen, 1997; Vallelonga and Svensson, 2014). Si is the second most
abundant element in the Earth´s crust (28.2%), forming silicates together with the most abundant element, O (46.1%). The





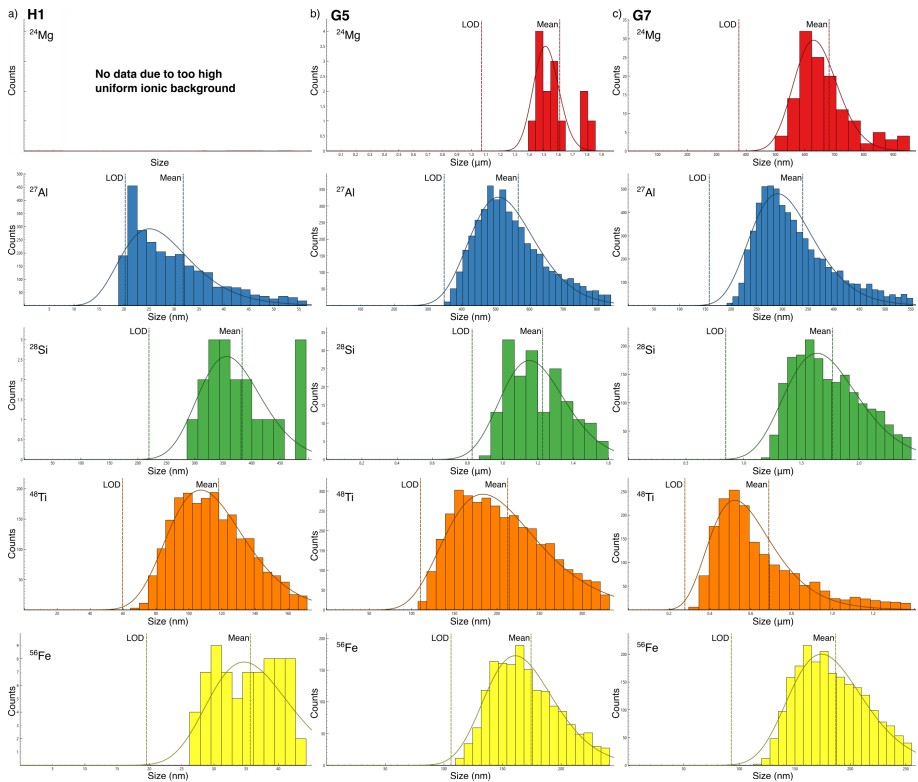

**Figure 5.** Particle size distribution for [24] Mg (red), [27] Al (blue), [28] Si (green), [48] Ti (orange), and [56] Fe (yellow) in a) H1, b) G5, and c) G7. Size estimates based on the chosen mineralogy as explained in Sect. 2.5. The histogram is estimated data based on measured data. The line indicates the lognormal distribution and should be treated with caution for the area below the limit of detection (LOD). LOD and mean values are indicated. Note $\mu$m on the x-axis for [24] Mg and [28] Si in G5 and [28] Si and [48] Ti in G7.

high concentration of [27] Al can be explained by Al being the third most abundant element of the crust (8.2%) and the most
abundant metal, forming several aluminium oxides and members of the clay, feldspar, and mica groups. The fourth most
abundant element in the crust is Fe (5.6%), forming iron oxides and being part of, for example, sulfides, mica, and feldspar
varieties, explaining the high concentration of [56] Fe (Fig. 3a). Especially in the glacial, dust minerals, such as mica, feldspar,
and hematite, are very abundant (e.g., Stoll et al., 2023b). The concentration and normalised shares of [48] Ti vary throughout our
samples as Ti is the ninth most abundant element (0.6%) in the crust, forming a variety of titanium oxides and other minerals.
[24] Mg shows the smallest concentrations and normalised shares in accordance with its relatively small concentration (2.3%) in
the Earth´s crust.

Our data show that the dust-rich cloudy band from the Younger Dryas (YD3) and the prominent cloudy band in G8 from
the last glacial (GS-12) have by far the highest concentrations (Fig. 3a). In both samples, [27] Al is the dominant element (Fig.
3b) followed by [56] Fe, [28] Si, and [48] Ti. [23] Mg-bearing particles always show the lowest numbers. As discussed by Stoll et al.
(2021a, 2023b); Bohleber et al. (2023), these cloudy bands are stratigraphic features indicating high insoluble particle concen-





trations partially from dry deposition events. These bands dominate the ice stratigraphy below the bubble-hydrate transition helping to asses the macroscale paleoclimate record integrity and are potentially areas of enhanced diffusion and microstructural ice rheology differences (Faria et al., 2010). The largest particles are also observed in YD3 and G8 (Fig. 4b). Only these two samples originate from colder periods, i.e. Stadials GS-1 (Younger Dryas) and GS-12. The mean particle sizes of all

elements, except $^{28}$Si, within YD3 and G8 are comparable, excluding H1 and G6 which contain finer-sized particles.

H1 is roughly 1000 years old and thus from a relatively warm period in the late Holocene, the Medieval Warm Period (e.g., Broecker, 2001; Mayewski et al., 2004; Goosse et al., 2006; Neukom et al., 2014). G6 is roughly 34.7 ka old and should thus also be from a Stadial, i.e. GS-7 (34.74-33.74 ka b2k) (Rasmussen et al., 2014). However, the low particle concentration and the small particle sizes (Fig. 3a and 4b) indicate that the age estimate or the cut sample is slightly off, which could have happened

during sample processing or cutting. The sample probably originates from the end of GI-7a (34.8-34.74 ka b2k) (Rasmussen et al., 2014), which is characterised by warmer temperatures.

The relatively small particles in H1 (Fig. 4b and 5a) and G6 (Fig. 4b) indicate a reduced insoluble particle transport to Greenland due to weaker winds during warmer periods. In contrast, aerosol transport was more potent during colder times, with stronger winds transporting larger particles farther, which can, thus, be deposited further from their source (Vallelonga

and Svensson, 2014) explaining the large $^{28}$Si-particles in YD3 and G8. Our limited sample range shows no direct evidence of proximal dust sources in warmer periods, as indicated by enormous particle sizes (above 8 $\mu$m) observed in the RECAP ice core (Simonsen et al., 2019).

We show that SP ICP-TOFMS yields new information helping in characterising climate periods based on particle concentration and size. Especially the analysis of nanometre-sized particles might show changes in the dominating wind systems that

remained hidden before. The proposed interstadial modes of atmospheric circulation brought relatively little dust to Greenland (Rasmussen et al., 2014), which is supported by our normalised particle number, concentration and size data. More data are needed for further in-depth characterisations. Investigating the dust particle size in the nanometre range could help calculate dust concentrations and the linking of future million-year-old ice core records, such as Beyond EPICA Oldest Ice, Million Year Old Ice or Dome Fuji Oldest Ice, with marine dust records from Antarctic Ocean Drilling Program (ODP) cores, such as 1090

and 1123 (Wolff et al., 2022).

### 4.2 Encouraging results from a multi-method investigation of a cloudy band

The presented data of YD3, the cloudy band from the Younger Dryas, show normalised shares of 19.5% and 20.4% Si- and Fe-bearing particles, respectively (Fig. 3b). Stoll et al. (2022) found 20% quartz ($SiO_2$) and 6% hematite ($Fe_2O_3$) in this sample. An in-depth investigation of an area within this sample with LA-ICP-MS 2D Imaging showed several areas interpreted

as clusters of heterogeneous Fe-, Si-, and Al-bearing particles (Bohleber et al., 2023). The authors report 20-28% "Si without Al" pixels, a quartz proxy, and 21% "Fe without Al" pixels, a hematite proxy. This comparison is encouraging, showing similar results except for the Raman spectroscopy-derived hematite share. However, obtaining representative statistics with confocal Raman spectroscopy is time-demanding and thus challenging, especially when focusing on one plane 500 $\mu$m below the sample's surface (Stoll et al., 2022, 2023b). Quantitative results are, thus, likely to differ, explaining this discrepancy. Another





important aspect is that the applied Raman spectroscopy set-up is limited to analysing micrometre-sized inclusions as they must be optically resolvable with a microscope. Large particles analysable with Raman spectroscopy are not represented in the SP data. The same holds for particles found with LA-ICP-MS, which is usually applied with a laser spot size of 10-40 $\mu$m. Further progress is expected with the untapped potential of LA-ICP-MS to measure with laser spot sizes of 1 $\mu$m.

This direct comparison exemplifies the potential for more systematic liaisons between different methods regarding the geochemical analysis of dust in polar ice. The applied methods, i.e., confocal Raman spectroscopy, LA-ICP-MS 2D Imaging and SP ICP-TOFMS, capture different types of signals, allowing snapshots of various characteristics. More data gathered with this multi-method approach might enable the development of enhanced generalisations of dust particles in the polar regions, deepening our understanding of the past climate.

### 4.3 Fe in particulate and dissolved form

One key element in the climate system is Fe as it can enhance the marine primary productivity, and is usually transported via aeolian mineral dust. We briefly discuss Fe and compare it with available ice core data. The overall mean proportion of $^{56}$Fe ionic background (with automatically chosen thresholds) ranges from 22.3% to 64.7% resulting in an average of 39% (Table 3). This value is slightly higher than the reported 27% for very young EGRIP ice by Erhardt et al. (2019). Fig. 6 shows our Fe/Al ratios, source region proxy, with age, including the mean value (1.13) of the continuous measurements conducted by Erhardt et al. (2019). The Fe/Al ratio for G1, the closest sample to the ice analysed by Erhardt et al. (2019), is 1.19 and thus very similar to the mean value reported by Erhardt et al. (2019) indicating a similar particle source region. Our samples are primarily within values reported for Asian dust (Jeong, 2008; Formenti et al., 2011) with the exceptions of G2, G5, and G6.

Differences in the Fe-proportions between this study and Erhardt et al. (2019) can be explained by the 1) methodological differences of both approaches and 2) different climate periods analysed. Our discrete samples give insights into different periods, i.e. the Holocene, Younger Dryas, and Glacial Stadials and Interstadials, over the last 50 kyr. In contrast, Erhardt et al. (2019) explore several continuous meters (roughly 86-93 m, i.e. 0.5-0.6 kyr b2k) from the Late Holocene focusing on implementing SP ICP-TOFMS into the Bern CFA system. We further focus on the merit of merging Raman spectroscopy, LA-ICP-MS 2D Imaging, and SP ICP-TOFMS to retrieve high-density data, enabling, for example, the estimate of particle sizes. Both approaches are complementary and we discuss the benefits and challenges of our approach in Sect. 4.2 and 4.6. The samples with Fe/Al ratios outside the Asian dust regime might support the recent argument that other regions than deserts in East Asia (e.g., Gobi, Taklamakan) and the Chinese Loess Plateau are source regions for dust deposited in Greenland (e.g., Újvári et al., 2022). Sr-Nd and Nd-Hf analyses of these samples are needed to follow up on this hypothesis.

### 4.4 Trace elements

Our data display the chemical variety of particles bearing numerous particulate elements ranging from very abundant, such as $^{28}$Si, $^{27}$Al, and $^{56}$Fe, to scarce (trace elements), such as $^{107}$Ag, $^{208}$Pb, and $^{118}$Sn (Fig. 2). Estimating mass fractions and mineral phases for trace elements is impossible due to lacking data and we thus only report the number of different particulate elements and respective detections.





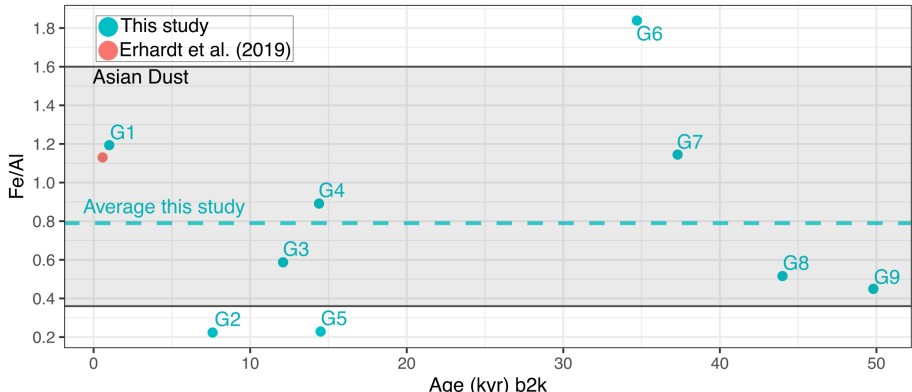

**Figure 6.** Fe/Al ratios in our samples (cyan), the mean value from Erhardt et al. (2019) (red), and dust values from Asia (Jeong, 2008; Formenti et al., 2011).

YD3 contains the highest number of different particulate elements (n=74), followed by G4 (n=73), G8 (n=71), and H1 (n=71). While YD3 is from the dusty Younger Dryas, H1 is only 1000 years old and from the Medieval Warm Period. There is evidence from the KCC ice core from Colle Gnifett and other records that there was increased dust production in Africa and that Saharan dust was transported more frequently to the Alps in the Medieval Warm Period (e.g., Bohleber et al., 2018; Clifford et al., 2019). This could explain the high number of particulate elements in H1, which could have originated from North Africa and Europe, as was recently shown for glacial ice (GS-3) from the NGRIP ice core (Újvári et al., 2022). However, as discussed above, the Fe/Al ratio of H1 fits typical ratios for Asian dust (e.g., Jeong, 2008; Formenti et al., 2011) (Fig. 6) contradicting a more diverse source region. A larger number of samples could help obtain better statistics on the number of particulate elements per sample.

Similar to H1, G6 also contains 70 different particulate elements despite their low particle concentrations (Fig. 3). Particles in G6 are also comparably small (Fig. 4b). This finding implies that the source regions and the respective source minerals of the insoluble particles within G6 may have been comparably diverse. Following the above-mentioned interpretation, this sample represents the climate transition from GI-7a to GS-7 (see Sect. 4.1). The particles inside the sample might originate from different regions depending on the prevailing wind systems during this period. Further analyses of this depth regime are needed to address these hypotheses.

There is little research on trace elements in snow and ice from Antarctica (e.g., Gabrielli et al., 2005) and Greenland (e.g., Barbante et al., 2003) and especially data from deep ice cores are missing. Glaciers from outside the polar regions have been investigated more closely regarding trace element concentration (Kutuzov et al., 2022); however, the main focus is on trace elements regarding anthropogenic pollution, and samples are thus comparably young. For example, low concentrations ($pg\,g^{-1}$) of Ag, Au, Pt, Pd and Rh were identified in ice and snow from the Mont Blanc in the French–Italian Alps with a maximum age of roughly 250 years (Van de Velde et al., 2000). In the Italian Dolomites, Gabrielli et al. (2008) identified several trace



elements, such as Ag, Ba, Cd, and Pb, in recent winter surface snow. For a long time, this lack of data was mainly due to the
missing sensitivity of the available analytical techniques and the extremely low concentrations of these elements in polar ice.

We present a proof-of-concept showing that trace element analysis on particles trapped in solid ice samples is possible
with SP ICP-TOFMS and could be the next step of systematic multi-method ice core analysis. An in-depth discussion of all
measured elements in our samples, their respective properties, and their potential implications is only feasible with more data
from EGRIP or other polar ice cores. This study may assist in sparking interest in a systematic investigation of trace elements
in deep polar ice cores utilising SP ICP-TOFMS, extending our understanding of, e.g., trace element fluctuations in the past.

## 4.5    Towards a more accurate representation of insoluble particle size in polar ice cores

Most particles in our samples are estimated to being in the nanometre size regime based on previous analysed with Raman
spectroscopy (Fig. 4b and 5). For some samples, the LOD of certain elements limits the exploration of the full particle size
distribution and small particles or particle with small elemental mass fractions may be missed. These findings are contrary to
most studies on insoluble particles in ice cores, which report a particle size distribution of around 1-2 $\mu$m (e.g., Petit et al.,
1981; Delmonte et al., 2002; Wegner et al., 2015; Potenza et al., 2016). However, already Steffensen (1997) showed that the
particle size distribution in the GRIP ice core was lognormal in the interval between 0.4-2 $\mu$m supporting the occurrence of
very small particles, which should be investigated further.

Established methods, such as the Coulter Counter and the laser particle detector, do not measure the entire dust particle size
range but are limited to certain size regimes and no compositional data are available (Simonsen et al., 2018). Coulter Counter
measurements are usually restricted to a size range between 0.6 and 10 $\mu$m, while the laser particle detector usually identifies
particles in a size range between 0.9 and 15 $\mu$m (Vallelonga and Svensson, 2014). Dust particle size is an essential constraint
for modelling atmospheric transport and the impact of dust on e.g., global radiation, clouds, and biogeochemistry but is still
misrepresented (e.g., Tegen and Lacis, 1996; Mahowald et al., 2014; Adebiyi and Kok, 2020). Past atmospheric modelling thus
also benefits from further in-depth size particle data from ice cores. Further SP analyses of ice could establish a more realistic
representation of insoluble particle size, especially in the nanometre range. Additionally, they can establish models on internal
and external mixing states and can therefore give size distributions for specific particle species.

Our approach to estimating particle size relies on a few assumptions regarding the mineralogy of the most abundant elements.
The Raman spectroscopy data previously gathered on the same samples (Stoll et al., 2022, 2023b) were supported by literature
values and are thus convincing assumptions. To further check this, instead of only using $Fe_2O_3$ (hematite) for [56] Fe particles,
we conducted tests using $Fe_3O_4$ (magnetite) as magnetite was identified in G8 by Stoll et al. (2023b). However, mean and
median particle sizes only decreased by ca. 2 nm for this configuration, supporting the validity of our Fe-bearing particle
size estimates. This could differ for other elements and respective minerals but is adequate for a proof of concept of a novel
approach to estimating particle sizes in ice core samples.





### 4.6 Challenges and limitations of ice-related SP analysis

The compositional analysis of particles can be divided into elemental and mineral. SP ICP-TOFMS can tackle the former, and counting sufficient particles allows to establish models on both the internal and external mixing states using, for example, statistical tools (e.g., hierarchical agglomerative clustering) (Tharaud et al., 2022). The mineral composition requires adequate spectroscopic techniques. Raman spectroscopy investigates light scattered inelastically by particles and accounts for only a fraction of the overall scattered light. As such, Raman spectroscopy is often limited to investigations of particles with dimensions at the microscale (depending on refractive index). However, surface-enhanced or stimulated Raman spectroscopy may be an interesting avenue to drive particle analysis in the future. A recent technique featuring an optical trap equipped with a single particle Raman module and linked to SP ICP-TOFMS may overcome current challenges and shows promise to characterize the same particle with different orthogonal techniques for accurate sizing, counting and compositional analyses (Neuper et al., 2024). Finally, the determination of particle number concentrations in ice core samples may be hampered by the large difference in the numbers of different particulate species especially when investigating a larger number of samples. For example, particles containing Fe, Al, Ti, or Si are very common, while other entities containing elements such as lanthanides or actinides are relatively rare. Choosing a uniform threshold for all elements for several samples enables inter-compatibility between samples but filters out a certain amount of particles. In H1, no data were available for $^{24}$Mg due to the high ionic background threshold originating from the dust-rich sample YD3 (Table A1). Additionally, artefacts could occur if elements are not analysed with consistent statistics when the automatic algorithm in SPCal is chosen. Depending on the focus of the analysis, it will have to be debated which approach is the most appropriate.

The calibration of number concentrations of abundant particles may require previous dilution, which may alter the matrix and consequently particle stability (Gonzalez de Vega et al., 2022). Investigating particles with low number concentrations, however, may require extended acquisition times. In summary, finding ideal conditions for various particles can be difficult, and a compromise is frequently required. In this study, analysis times were 100 s, and no dilution was carried out to conserve natural ionic strength. Especially in particle-rich ice, such as cloudy bands, some particles were so abundant that coincidental detection was likely. The applied threshold method should be a good approach to tackle this issue. However, it is worth noting that at higher coincidences, calculating accurate number concentrations, sizes, and compositions becomes increasingly difficult, and numbers should only be regarded qualitatively as particle numbers may be significantly underestimated while sizes are increasingly overestimated. SP ICP-TOFMS is an excellent tool to study particles across the nanoscale. However, it is worth noting that at high sizes (e.g., in the micrometre scale) ionisation of particles may be incomplete or detector saturation may be reached. This needs to be evaluated on a per-case basis to ensure accurate characterisation of distributions of large particles.

### 4.7 Implications

This study 1) shows the potential of applying a holistic multi-method approach regarding impurities in ice and 2) advances the knowledge about particles, especially in the sub-micrometre size range, setting a new benchmark for size and elemental



and mineralogical composition studies on dust in ice. We do not aim to derive seasonal signals, as possible with SP analysis coupled to CFA (Erhardt et al., 2019), but this might also be possible within certain boundaries. Analysing a continuous set of

small, discrete samples from depths without strong layer thinning could resolve quasi-annual signals and remains to be tested. We exhibit that such an approach is possible with small ice volumes, such as the used cubic samples with dimensions of often less than 1x1x1 cm.

We tapped into the largely unexploited potential of SP for ice core analyses, especially in combination with other state-of-the-art methods. In the wake of projects derived by the International Partnership in Ice Core Sciences (IPICS) aiming to

recover the "Oldest Ice", retrieving as much information as possible from each sample, especially from the deepest, highly thinned regions, is essential. Alongside the need for a high spatial resolution to resolve these layers, other areas of interest are the "geochemical reactor in ice" hypothesis (Eichler et al., 2019; Baccolo et al., 2021) and the investigation of dust particle size and clustering. Bohleber et al. (2023) and Stoll et al. (2021a, 2022) found clusters of insoluble particles, which likely separate during the melting process of CFA. Analysing the chemical co-localisation of particles with SP ICP-TOFMS data has

the potential to help explore this issue. Following up on work by Simonsen et al. (2019), analysing RECAP ice, especially from the glacial, with SP ICP-TOFMS, would be an attractive target to explore ice sheet retreats and advances. Furthermore, investigating dust sizes in million-year-old ice could illuminate potentially existing local dust sources in the terrestrial margins of Antarctica during the Mid-Pleistocene Transition (Raymo et al., 2006; Wolff et al., 2022). Including SP ICP-TOFMS in ice core analysis routines will immensely help when details on dust characteristics are needed. Developing a "best practice"

approach for SP ice core analyses is necessary to foster comparability of results coming from different labs. This ranges from laboratory guidelines, such as decontamination procedures, to coordinated data processing and analysis routines via dedicated software or in-house scripts. For the presented particle size estimate, it would be beneficial to conduct dedicated overview measurements with Raman spectroscopy on, e.g., a few samples per climate period, delivering benchmark minerals as input parameters. Taking advantage of the state-of-the-art in analytical chemistry could further push comprehensive particle

characterisation by applying, e.g., online hyphenation of optofluidic force induction (OF2i) coupled with Raman spectroscopy and ICP-TOFMS (Neuper et al., 2024). This approach would enable the trapping of particles in a vortex beam and, thus, their direct characterisation via Raman spectroscopy before being analysed by SP ICP-TOFMS. The spatial information regarding the microstructural localisation of the particles would be lost. However, the assumptions taken regarding particle mineralogy for the particle size estimates would be more robust than in our pilot study.

The approach of utilising the strengths of various microstructural and impurity methods on the same samples (Stoll et al., 2021a, 2022; Bohleber et al., 2023; Stoll et al., 2023b), has eventually ended with the application of SP ICP-TOFMS and the melting of the respective samples for this study. With careful planning, dedicated ice samples could be analysed efficiently by integrating a cascade of analytical methods, eventually ending with SP analysis and potentially being compared to parallel CFA measurements on an adjacent piece of ice and 3D modelling (Larkman et al., 2024). In addition to the knowledge gained,

such a strategy would strengthen interdisciplinary and international collaborations among ice-core researchers and inorganic chemists.





# 5 Conclusions

SP ICP-TOFMS analysis has excellent potential for impurity-related ice core research. We show the first single particle inductively coupled plasma-time of flight mass spectrometry data from a polar ice core covering several climate stages, such as the late and intermediate Holocene, the Younger Dryas, and different Glacial Stadials and Interstadials. We observe substantial differences between the nine samples in normalised particle number, concentration, and dominant composition. We focus on some of the most abundant elements in the Earth´s crust, i.e. $^{27}$Al, $^{56}$Fe, $^{28}$Si, $^{24}$Mg, and $^{48}$Ti. We further introduce a new technique to estimate the size of each particle based on its measured elemental chemistry and presumed mineralogy based on existing Raman spectroscopy data. Particles range from a few nanometers to micrometres and thus give new in-depth insights into the size distribution of dust particles transported to Greenland. This study presents the advanced stage of a systematic multi-method analysis approach for dedicated ice core samples, merging the benefits of Raman spectroscopy, laser-ablation inductively coupled plasma mass spectrometry, and SP ICP-TOFMS measurements. Incorporating such an approach in the planned analyses of precious million-year-old ice would foster interdisciplinary collaborations and boost the scientific outcome.

*Data availability.* Data will be publicly available on PANGAEA as soon as the manuscript has been accepted.

*Author contributions.* NS wrote the initial manuscript with contributions from all co-authors. Samples were prepared by NS and measurements were conducted by NS, DC, QGV, PL, and PB. Data processing was done by NS, DC, ME, and RGV. Funding for NS and PB was acquired by PB.

*Competing interests.* No competing interests exist.

*Acknowledgements.* Nicolas Stoll and Pascal Bohleber gratefully acknowledge funding by the Programma di Ricerca in Artico (PRA). This work was further supported by Chronologies for Polar Paleoclimate Archives – Italian-German Partnership (PAIGE) and the "Initiative and Networking Fund of the Helmholtz Association". Pascal Bohleber gratefully acknowledges funding from the European Union's Horizon 2020 research and innovation program under the Marie Skłodowska-Curie grant agreement no. 101018266 and further funding by the European Union (ERC, AiCE, 101088125). Views and opinions expressed are however those of the authors only and do not necessarily reflect those of the European Union or the European Research Council. Neither the European Union nor the granting authority can be held responsible for them. Piers Larkman gratefully acknowledges funding from the European Union's Horizon 2020 research and innovation programme under the Marie Sklodowska-Curie grant agreement no. 955750. We thank Tobias Erhardt, Hubertus Fischer, and Geunwoo Lee for fruitful discussions. EastGRIP is directed and organised by the Centre for Ice and Climate at the Niels Bohr Institute, University of Copenhagen. It is supported by funding agencies and institutions in Denmark (A. P. Møller Foundation, University of Copenhagen),



USA (US National Science Foundation, Office of Polar Programs), Germany (Alfred Wegener Institute, Helmholtz Centre for Polar and Marine Research), Japan (National Institute of Polar Research and Arctic Challenge for Sustainability), Norway (University of Bergen and Trond Mohn Foundation), Switzerland (Swiss National Science Foundation), France (French Polar Institute Paul-Emile Victor, Institute for Geosciences and Environmental research), Canada (University of Manitoba) and China (Chinese Academy of Sciences and Beijing Normal University).



**Appendix A**

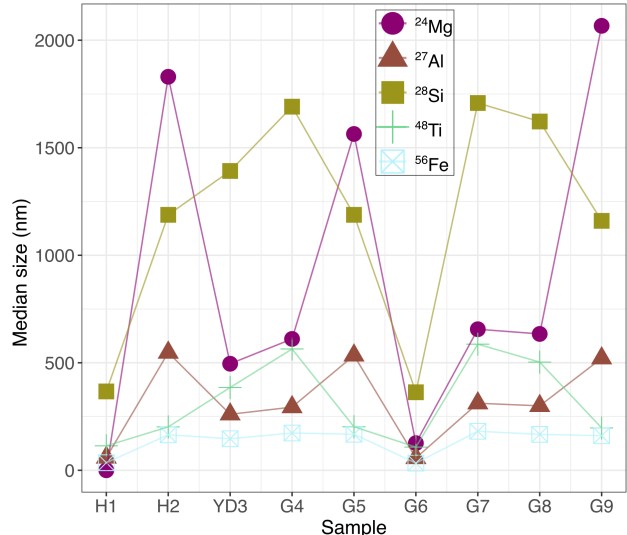

**Figure A1.** Median size estimates with uniform ionic background thresholds for $^{24}$Mg, $^{27}$Al, $^{28}$Si, $^{48}$Ti and $^{56}$Fe. Size estimates are based on the chosen mineralogy as explained in Sect. 2.5.





**Table A1.** Calculated ionic background thresholds for all samples and elements (part 1). Bold text refers to the highest ionic background threshold for the respective element applied to all elements for the size estimate.

| Sample | Chemical species | Calculated ionic background |
|--------|------------------|------------------------------|
| H1 | $^{24}$Mg | 8.979 |
| H1 | $^{27}$Al | 8.135 |
| H1 | $^{28}$Si | 8.474 |
| H1 | $^{48}$Ti | 8.593 |
| H1 | $^{56}$Fe | 10.73 |
| H2 | $^{24}$Mg | 7.135 |
| H2 | $^{27}$Al | 9.018 |
| H2 | $^{28}$Si | 8.999 |
| H2 | $^{48}$Ti | 7.344 |
| H2 | $^{56}$Fe | 20.2 |
| YD3 | $^{24}$**Mg** | **68.75** |
| YD3 | $^{27}$Al | 38.81 |
| YD3 | $^{28}$**Si** | **308.1** |
| YD3 | $^{48}$**Ti** | **35.38** |
| YD3 | $^{56}$**Fe** | **424.5** |
| G4 | $^{24}$Mg | 48.25 |
| G4 | $^{27}$Al | 13.87 |
| G4 | $^{28}$Si | 26.06 |
| G4 | $^{48}$Ti | 24.78 |
| G4 | $^{56}$Fe | 20.2 |
| G5 | $^{24}$Mg | 8.348 |
| G5 | $^{27}$Al | 12.59 |
| G5 | $^{28}$Si | 11.77 |
| G5 | $^{48}$Ti | 9.018 |
| G5 | $^{56}$Fe | 104 |
| G6 | $^{24}$Mg | 17.13 |
| G6 | $^{27}$Al | 14.9 |
| G6 | $^{28}$Si | 18.34 |
| G6 | $^{48}$Ti | 15.27 |
| G6 | $^{56}$Fe | 25.27 |



**Table A2.** Calculated ionic background thresholds for all samples and elements (part 2). Bold text refers to the highest ionic background threshold for the respective element applied to all elements for the size estimate.

| Sample | Chemical species | Calculated ionic background |
|--------|------------------|------------------------------|
| G7 | $^{24}$Mg | 20.34 |
| G7 | $^{27}$Al | 14.26 |
| G7 | $^{28}$Si | 19.76 |
| G7 | $^{48}$Ti | 14.74 |
| G7 | $^{56}$Fe | 23.93 |
| G8 | $^{24}$Mg | 44.84 |
| G8 | $^{27}$**Al** | **40.78** |
| G8 | $^{28}$Si | 211.8 |
| G8 | $^{48}$Ti | 23.42 |
| G8 | $^{56}$Fe | 393.5 |
| G9 | $^{24}$Mg | 7.398 |
| G9 | $^{27}$Al | 7.502 |
| G9 | $^{28}$Si | 8.87 |
| G9 | $^{48}$Ti | 7.004 |
| G9 | $^{56}$Fe | 9.803 |

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
