# Peer review of "Single particle ICP-TOFMS on previously characterised EGRIP ice core samples: new approaches, limitations, and challenges"

_EGUsphere, 2025_

## Author Comment (AC1)

R1: I have read this manuscript with great interest, particularly the geochemical analysis using elemental data from sp-ICP-TOFMS, which complements the authors' previous research. I fully agree that ice core samples should be analysed using multiple techniques to extract climatic signals from polar ice from different perspectives. Since no single method can capture the complete climatic record stored in ice, this study serves as an excellent example of utilizing multiple analytical approaches for dust analysis in the same Greenland ice cubes. This comprehensive investigation can provide valuable multi-dimensional insights into Earth's climate variability.

However, upon closely examining the sp-ICP-TOFMS measurement results, I was highly concerned about the data quality. This raises substantial concerns regarding potential contamination of the ice cube samples used in this study. Below, I outline the reasons for my concerns regarding data quality related to sample contamination. Until the authors address these contamination issues clearly, I am unable to provide a complete review of this manuscript.

A: We thank the reviewer for accepting this review and are grateful for the comments. However, we respectfully disagree with the reviewer, in particular regarding the data quality and decontamination. We can address the comments as we outline below. In doing so, we hope to improve the manuscript´s quality and avoid misunderstandings. Below, we will reply in detail to the raised issues. We quote the reviewer here to emphasise the objectives of this study:" I fully agree that ice core samples should be analysed using multiple techniques to extract climatic signals from polar ice from different perspectives. Since no single method can capture the complete climatic record stored in ice, this study serves as an excellent example of utilising multiple analytical approaches for dust analysis in the same Greenland ice cubes." In summary, we aim to display a multi-method approach to analyse ice samples using several methods; only the final stage contains single particle analysis, but we demonstrate here that this is possible at no expense of the data quality. This objective should be considered when discussing the data. We do not aim to produce a stand-alone data set comparable to a CFA record but to compile complementary data from sophisticated analyses on one sample set. For this purpose, ice core samples dedicated to CFA analysis cannot be used because they are destroyed by the melting of the samples. Thus, other approaches, including using different core parts, must be explored - especially for analysing precious, highly thinned layers of million-year-old ice.

R1: **Potential contamination issues**
The methodology employed in this study is novel and has not been previously established. Given that trace metal analysis of polar ice cores requires extraordinary precautions to prevent contamination, additional examination is necessary. The authors used the "physical property" sections of the EGRIP ice core—these outermost sections are typically used for physical property measurements, where contamination from external materials (e.g., drilling fluid) is less critical. However, for trace metal analysis, contamination risk must be minimized.

The manuscript describes a decontamination process, but given the dimensions of the ice cube samples (1 cm × 1 cm × 1 cm, 1 mL of ice), I am concerned that this procedure is insufficient to effectively eliminate contamination. Established decontamination protocols for ice cores retrieved from fluid-filled boreholes require an initial acetone rinse to remove drilling fluid, followed by ultrapure water rinsing until 20–60% of the original ice volume has melted (Boutron and Batifol, 1985; Delmonte et al., 2002; Gaspari et al., 2006). The authors should provide a more detailed description of their decontamination procedure and critically assess its suitability for trace element analysis in polar ice cores.

A: We are unsure which methodology is addressed here: decontamination or single particle analysis. In particular regarding the second issue, we respectfully disagree. The decontamination of discrete samples with MQ water (e.g., Delmonte et al. 2004) under an air flow bench using gloves is an established technique, e.g., in dust particle research using the coulter counter. We further applied all the knowledge and experience of our co-authors, analytical chemists working on nano- and single particles, who are well-trained in ultra-trace analysis, in the lab. The used samples are always from the inward-facing side of the physical properties (PP) piece at its thickest location and thus never had contact with the outside of the core or the drilling liquid. For the previously conducted studies on these samples, it was essential to use intact samples containing no micro-cracks, which was ensured by microstructural analyses utilising Large Area Scanning Macroscopy and Microstructure Mapping (Stoll et al., 2021, 2022, 2023). This makes it highly unlikely that drill liquid has reached our samples. If this is assumed, the adjacent surface of the CFA and gas piece would also be contaminated (see figure below). Initial sample decontamination included several steps of microtoming for

Raman spectroscopy and later decontamination with ceramic knives for laser ablation measurements. Thoroughly rinsing of the sample with MQ water from all sides and melting the outer layers was the final step. Thus, the initial ice sample volumes were melted to at least 60%, which is in agreement or higher with the studies mentioned by the reviewer. It should further be acknowledged that drill liquid quality has improved over the last decades, and the EGRIP driller used state-of-the-art drill fluid (M. Hüther, pers. communication 03.03.25). Additionally, the analyses conducted before the SP-ICP-MS analysis (cited in the manuscript) did not show unexpectedly high values of any element analysed, supporting that no initial contamination took place or occurred due to sample storage and handling.
We suggest to provide a more detailed description of the decontamination method in the revised version. We would add additional text regarding the potential risks of the chosen samples, highlighting that using other ice core pieces, such as CFA, would be preferable. Unfortunately, this remains unlikely due to the limited availability of samples, but it could spark meaningful discussions and collaborations in the community for future studies.

[Figure]

*Figure 1: Cutting scheme of the EGRIP ice core. The red box displays the approximate position of the used samples overlapping with CFA and Gas pieces.*

R1: Concerns about sample cleanliness become even more pronounced when comparing the ionic Fe background thresholds in Table A1 with total Fe levels reported in two well-aligned Greenland ice core studies: North Greenland Eemian Ice Drilling (NEEM) and EGRIP (Burger et al., 2021; Erhardt et al., 2019).

Typically, total Fe concentrations (including both particulate and dissolved Fe) should be significantly higher than ionic Fe background thresholds. However, the reported values in this study appear unexpectedly high.

First, the unit for ionic Fe background thresholds is missing from Table A1. Assuming it is in ppb (ng/g), which is a common convention, the lowest and highest Fe ionic backgrounds in this study are 9.803 ppb and 424.5 ppb for the cold climate samples G9 and YD3, respectively. These values are alarmingly high compared to the total dissolvable Fe concentrations of 2.9–146.4 ppb reported by Burger et al. (2021) after a month-long acidification process. Despite differences in sampling locations within Greenland, such high background thresholds may suggest contamination.

Furthermore, dissolved Fe background levels in a 3-m EGRIP ice core from the Holocene typically range from sub-ppb to 6 ppb, with non-dusty sections near 0 ppb and dusty sections around 2–3 ppb (Erhardt et al. 2019). The authors attribute the discrepancy between these two studies to methodological differences and different climate periods (line 303), but given that both Holocene EGRIP ice core samples exhibit similar dust particle concentrations (~1000 particles/mL, as measured by a laser particle counter (Abakus)), the observed Fe background levels in this study remain questionable.

A: This comment is based on a misunderstanding, which we hope to clarify here. The mentioned thresholds are applied to make quantitative sample comparison possible, which is a big challenge in single particle analysis of samples with vastly different concentrations, as found in, e.g., Holocene and Younger Dryas ice. Thresholds are defined based on a statistical analysis of the data set, in which the mean value of the data set is considered and compound Poisson statistics are applied. These statistics are used to determine an alpha value of $10^{-6}$ (meaning a false positive detection rate of 1 in a million detection events), which provides the "threshold" over which a detection event is considered a particle signals. As such, the threshold is a statistically derived parameter for pinpointing particulate signals and does not provide quantifiable particle parameters besides the size detection limit. This threshold only influenced by the sample mean value and therefore, when comparing data sets with different ionic Fe levels, thresholds change. This results in different size detection limits which complicates the comparison of numbers and sizes of found particles across a sample set. To enable a consistent comparison, we used a conservative measure, in which the highest threshold in a sample set was used across all other samples to enable a comparison. More detailed information on this procedure can be found in the cited references. In summary, the thresholds do not reflect mean Fe concentrations. However, we would like to point out that this study specifically chose dust-rich samples based on CFA and visual stratigraphy data (Stoll et al., 2021, 2022, 2023), it is thus expected that these samples have comparably high concentrations.

The comparison to the Fe values obtained by Burgay et al. (2021) is inept. As displayed in section S1 and table S2 (Burgay et al., 2021), the authors analyse four small parts of a very shallow (100.8 m) and, thus, very young (maximum age: 1363 CE, oldest analysed sample 1611-1596 CE) core. Derived data is thus not comparable to our samples from the Younger Dryas or glacial often containing cloudy bands and thus a much higher insoluble particle content. On the opposite, this strengthen the motivation of our study as we explicitly mention that data is challenging to interpret as so little data from deep ice cores is available resulting in the need for more single particle studies on ice.

We suggest enhancing the clarity of the text in a revised version to avoid further misunderstandings regarding the applied thresholds. We will make sure to clarify that the applied thresholds are intensity values (in counts) and not concentrations.

Below is a figure displaying the thresholds and the ultra-pure blanks measured in the same analysis runs. The black line indicates the compound Poisson threshold (critical limit for a=$10^{-6}$) for the respective element in the blank. The dashed blue line indicates the highest threshold across all samples (taken from tables A1 and A2), which we used for all data evaluation.

Both combined display that our sample handling, ultra-pure water and the instrument itself do not create any particle artefacts for the chosen thresholds, which are comfortably high. There are zero data points above the dashed line, meaning we virtually do not expect any false positive events from

the "instrument/chemical" blank.

[Figure]

Figure 2: Comparison of ultra-pure blanks (black) and chosen thresholds (blue dashed line). No contamination is visible.

In addition to the high Fe background levels, the inconsistency in particle counts and sizes for the five major elements in Figure 5 further suggests a high probability of contamination. Since Mg, Al, Si, Ti, and Fe primarily originate from mineral dust, their dissolved and insoluble concentrations should be positively correlated unless there is a specific event providing an element only without the others. For instance, if a sample exhibits a high dissolved Fe background, a correspondingly high Fe particle concentration would be expected. Similarly, increased dust content should be reflected in Al and Mg concentrations due to their co-occurrence in mineral dust, and vice versa. However, the observed discrepancies in detection trends—for example, high Al but low Fe detections in sample H1 compared to samples G5 and G7—suggest inconsistencies in the data that are difficult to be produced with natural sources alone.

A: The difference in particle counts and sizes displayed in Fig. 5 is due to the applied thresholds using the highest respective thresholds to enable the quantitative comparison between samples with very different particle concentrations. For example, the high threshold for [56]Fe derived from YD3 (Table A1) is applied to all other samples (while the [27]Al threshold is much lower) thus explaining the difference in the number counts in Fig. 5 (see also previous comment).

The use of physical property sections—directly exposed to drilling fluid, uncleaned processing and packaging materials—introduces a high risk of contamination, making these samples unsuitable for trace-level analyses without strict decontamination procedure. Additionally, given that these samples have been handled in different environments over the years for multiple analyses, the risk of contamination is further increased. Due to their small size, effectively decontaminating 1 cm³ ice samples for trace metal analysis is particularly challenging.

A: As explained above, we disagree with these statements. The used samples were never directly exposed to drilling fluid. Further, the processing and packaging in the field procedures were the same for all EGRIP ice core pieces. We can assure you this as the lead author has been involved in four EGRIP field seasons and leading the science trench twice. We conducted strict decontamination procedures at the University of Graz under the supervision of established analytical chemists. Handling and decontaminating small ice samples is challenging but was successfully done, as displayed by the ultra-pure blanks, which we handled likewise. In fact, it should be considered an important result of this work that such sequential multi-method studies are possible, even when highly-sophisticated analyses are involved. In addition, spICP-TOFMS analysis works well

for small sample volumes of less than 500 μL while enabling analysis of all chemical elements. We will extend the text to stress this point. At the same time, we see greater risk for introducing biases to the SP-ICP-MS results if samples are stored in vials for prolonged periods, refrozen, or exposed to even mild acids. Based on or our approach we have successfully avoided all of these potential pitfalls.

This proof-of-concept study displays the idea of several cascading measurements enabling as much data as possible from the same sample. In future studies, starting with samples of larger volume would be beneficial to make decontamination easier. However, this has the drawback that analyses take longer to achieve representative results, especially for time-consuming analyses such as confocal cryo-Raman spectroscopy, which must be focused on every single particle inside the ice sample. The recent progress in analytical techniques and multi-method analyses of ice core samples and the needed strengthening of a holistic microstructural and geochemical analysis evokes rethinking and updating historically grown ice core sample routines, which might not apply to the same degree any more.

**Conclusion**

The study presents a good case for multi-technique ice core analysis. However, the potential contamination issues outlined above need to be thoroughly addressed. The authors should provide clearer details on their decontamination process, re-evaluate its effectiveness, and discuss how contamination might have influenced their results. Without these clarifications, the reliability of the data remains uncertain, preventing further evaluation of the manuscript.

A: We thank you again for this review and hope we could clarify the addressed points of criticism. We are happy to include more details on the decontamination, the specific origin of the sample, and previous contamination steps. We will further discuss the advantages and disadvantages of the used samples. These clarifications will help focusing on the main objectives of this study, the presentation of a novel multi-method approach and the merit of single particle analyses on ice cores for a vast field of research fields.

References
Delmonte, B., I. Basile-Doelsch, J-R. Petit, V. Maggi, Marie Revel-Rolland, A. Michard, E. Jagoutz, and F. Grousset. "Comparing the Epica and Vostok dust records during the last 220,000 years: stratigraphical correlation and provenance in glacial periods." *Earth-Science Reviews* 66, no. 1-2 (2004): 63-87.

---

## Author Comment (AC2)

Stoll and co-authors present results from single particle time-of-flight (SP TOF) ICP-MS analysis on EGRIP (Greenland) ice core samples that have already been analysed by Raman spectroscopy and laser ablation (LA)ICP-MS. Overall, I find the subject very interesting and would like to see publication. Before this, I recommend that the authors consider significant revision to ensure the case they attempt to present for this new method is a compelling one, based on robust interpretation of the data, accounting for assumptions and uncertainties.

A: We thank the referee for their helpful comments which will improve the manuscript significantly. We are happy to include more details and the suggested changes in a revised version. Below, we will answer the raised issues in blue.

I found the manuscript difficult to read. After reading the abstract, introduction, and conclusions (what many readers initially do) I was no wiser as to the specific focus of the study. The abstract includes several nebulous claims of 'complimentary perspectives' and 'new possibilities' but it is not clear what these are. From reading the manuscript through, the only advantage of including SP TOF analysis in the chain of particle-related analytical methods presented by manuscript is the chance to estimate particle size distribution from the SP TOF results. It would be better if the abstract, conclusions, and likely also the title, focused on this aspect, rather than claiming any benefits that are not demonstrated in this study.

A: We will tighten the writing to provide a more streamlined reading experience. As described in the manuscript, the SP technique is so far hardly used in ice core sciences despite its benefits. These go far beyond the aspects mentioned in the above comment and we will highlight critical advantages within the revision. In brief, with SP ICP-TOFMS, it is possible to gain single particle resolution from an elemental perspective. Especially the TOF mechanism is a game changer as it enables non-target screenings for particulate elements without a priori information. Found particle signals carry convoluted information which enables us to calibrate number concentrations for selectable particle populations, to determine mass distributions and estimate sizes, as well as to carry out an in-depth composition analysis which provides opportunities to identify geochemical signatures. As such, several critical parameters for particle analysis can be retrieved, which is not possible with current techniques. However, it is fair to point out that the purely "elemental" perspective is limited and single particle Raman spectroscopy provides a unique complementary vision enabling to withdraw mineral data enabling a much better modelling of particle composition and adding information which otherwise remain hidden. We will highlight these advantages with more emphasis in our revision. This emphasis was weaved into the manuscript in a way, which also makes the aims and objectives easier to identify.
The presented approach is a potential systematic analysis tool for the future especially for very rare samples from the "Oldest Ice" quest, such as Beyond EPICA, Dome Fuji or Blue Ice samples.

Why develop a new method for particle sizing at all? Why not stick with CFA-based Abakus, SPES or Coulter counter methods? It would be great to see some justification of the need for the method development proposed here. L285-288 could be moved from the Discussion for example.

A: We mention that the commonly applied techniques (Coulter Counter and Abakus) have limitations, such as the measurable particle size range (usually above 1 micron, l. 160). SPES is indeed exciting progress for nanoparticle size measurements (particle minimum size of 0.2 micron in Zeppenfeld et al. 2024) and we will include SPES in the revised text. However, SP ICP-TOFMS provides much deeper insights into ice cores and accesses several particulate parameters simultaneously at single particle resolution. It becomes possible to detect one particle and determine mass/size as well as its elemental composition. Importantly, we may detect particles as small as 30-50 nm, which goes far beyond the capabilities of the aforementioned techniques. Through the counting of thousands of particles per minute, we gain the ability to distinguish between different (small!) particle populations which have different significance regarding their presence in ice and through clustering methods, we may estimate their identity. Through the consideration of cluster-specific numbers, mass/size distributions as well as faint chemical impurities, we gain insights, which remain hidden when only using established methods.

Thus, we believe that it is advantageous to explore such new approaches enabling a more holistic view into particle characteristics. Particle size is important for e.g. climate modelling while both, chemical composition and particle size, are important to explore the role of dust during the Mid-Pleistocene Transition (Wolff et al., 2022). We will focus more strongly on these aspects in the revised version and move important information to the introduction as suggested.

A couple of technical queries:

- Could more information be provided on the conversion from measured 'intensity' (Fig. 1) to a 'detection' (is this the same unit as intensity, minus the threshold value?), to 'normalised detection' (normalised to what?).
  A: The exact workflow has been published previously, and detailed information on raw data streamlining is available in Lockwood et al. (2021, 2024).
  In brief: We use compound Poisson statistics to establish a threshold over which a signal is identified as a particle event with a certainty of 99.999%. Each signal is usually resolved across 3-10 data points, which are summed up as "detection" (or one particle event) and the same time, we automatically check for coinciding signals from other elements, which will be associated as "present within the same detection/particle event". The mean signal is subsequently subtracted and processed data is saved as an array for subsequent calibration, in which the elemental response and the transport efficiency is used to determine masses and sizes. Subsequently, we use hierarchical agglomerative clustering to suggest common particle populations. As described in the text and the answer to referee 1, different background thresholds have to be applied depending on the data analysis approach and the measured samples. To compare samples with highly different dust concentrations, we applied the same background threshold thus influencing the number of detections. Normalised detections are relative values referring to the percentage of a certain element found in the analysed sample. This enables a qualitative comparison between samples with highly varying particle concentrations, as is the case for our samples, ranging from the Holocene to the Younger Dryas and Last Glacial.

- L43 "if suitable standards are analysed concurrently". What are the standards referred to here? Are these the "calibration standards" mentioned at L107? How do you calibrate for ionic species and insoluble (particulate) elements? Or (related to above point) does a true calibration not actually occur?
A: There are two critical parameters in SP ICPMS, which need to be calibrated with a set of two standards: 1) A particle standard containing a particle with known composition, size and density (or alternatively with a known number concentration), which is analysed to determine the transport efficiency (see answer below). 2) An ionic standard (as well as a blank for background subtraction) containing all elements to be calibrated at a known concentration, which is analysed to determine elemental responses and to calibrate raw intensities into masses. If mineralogy data is available (which is the case if Raman spectroscopy is carried out in tandem), we can further translate single particle masses into sizes considering phase density and elemental mass fractions. The streamlined and automated process is explained in much detail in our previous (and cited) publications: Lockwood et al. (2021) and Lockwood (2024).

- How representative do you expect the Au nanoparticle recovery result to be of particulate matter within these ice core samples?
A: The Au nanoparticles are not used as internal standard. They are used to determine the transport efficiency in SP ICP-MS (see answer above). This efficiency describes the fraction of liquid nebulized and transported into the plasma. Using conventional set-ups, this efficiency is only around 5%, which means that from 100 particles, we only see 5. When attempting for example calibrations of number concentrations, this parameter is applied to consider the remaining 95%.

How well is the efficacy of the proposed SP TOF method for particle sizing demonstrated…?
A: SP ICP-TOFMS is a mass sensitive technique. As such we can accurately determine elemental masses within a single particle, which typically are within the ag to pg range. The sizing efficacy is dependent on complementary knowledge and also on some assumptions. On the one hand, we require some information on mineralogy to estimate particle density and mass fraction to translate mass into size. Especially in environmental studies, mineralogy is usually based on a sophisticated guess. However here, instead of guessing mineralogy, we determined mineral composition using Raman spectroscopy, which enables us to have more accurate models of size distributions making our study innovative. On the other hand, no knowledge on particle shape is available and SP ICP-TOFMS is used to project masses into a spherical shape. Admittingly, this does not reflect reality well but provides an estimate on the size scale of particles. Similar assumptions and limitations are present in other particle size analysis devices (e.g. Simonsen et al. 2018).

The authors state "Estimating particle sizes is possible if specific crystal phases are chosen for each element to obtain phase density and element mass fractions." In section 2.5 they describe how each element measured is assigned a mineral, based on previously published Raman spectroscopy work. This assumes, for example, that all Si is sourced from SiO2, all Al from potassium feldspar, and that the mineralogy present in the nanoscale particles matches that of the microscale particles

measured. These both seem like huge assumptions. The authors state the mineralogy assignment is a "simplification". I don't see that the implications of these assumptions are tested, i.e., to what extent do they influence the particle size results obtained? Si and Al are, by definition, present in all aluminosilicate minerals, which have different densities and elemental fractions.

A: It is right, that sizing is based on various assumptions as outlined in the previous answer. SP ICP-TOFMS is more applicable to detect element masses within a particle. However, this data is difficult to interpret and size information (e.g., 50 nm), provides a better understanding than mass data (Fe mass: 50 fg). As such, it is common to give a size estimate and admittingly, this sizing is subject to various assumptions and errors. In the past, environmental studies entirely guess mineralogy and while out approach is not flawless, we have some understanding of common mineral phases based on hundreds of previously characterized particles in the same samples. However, we admit that this still provides inaccuracies, because not all Al and Si are present as feldspar and a significant fraction is likely to consist of aluminosilicates. Nevertheless, our approach is still more evidence based than the current state of the art. Overall, we still think that our approach is a sensitive compromise of assumptions and available data. This could change if a coupled SP-Raman system trapping particles could be used (Neuper et al. 2024) providing direct in-situ data, but this has not been achieved yet for cryogenic analysis.
We will emphasize in the revised manuscript that our size estimation is prone to several assumption and simplifications, and point out that the aim is to provide a general understanding of potential size distributions without claiming they are absolute.

In section 3.2.1, Figure 4, each element (or isotope) has been assigned to a mineral, and each mineral is assumed to have a certain density. Are these provided anywhere? – this choice seems absolutely critical to the particle size estimate, if I understand correctly. Overall, the conversion from mass to size and the potential uncertainty is not clear – the reader is referred back to Section 2.5, which provides little help.

A: This is true, the chosen values are currently only indirectly mentioned via the mineral phases. We will provide the used densities in the revised text. See answers above for the mass-size approach as well as the mentioned references describing the conversion.

Figure 5 displays the calculated particle size distributions for the five chosen elements/mineralogy assumptions for three of the samples only. The axes labels are impossible to read so it is difficult to begin to judge how these distributions might compare to existing particle size data.

A: *TC* only allows a figure maximum width of 12 cm limiting the number of displayed examples without making the plots completely unreadable. Thus, mean and median values are displayed separately. We will enlarge the labels and update the plot to enhance readability.

Only samples H1, H2 and YD3 have insoluble particle data available (and only H1 is plotted on Fig 5). There doesn't appear to be any comparison with these existing

data within the manuscript. Unless I missed it, there is no attempt to verify the results of the SP TOF particle sizing method using independent means.

A: Unfortunately, insoluble particle data for the EGRIP ice core is limited as it is primarily motivated to gain a better understanding of ice flow and deformation. Thus, only specific depth regimes of the core have been analysed with CFA and large parts of the core are not planned to be analysed at the moment. Insoluble particle concentrations for the upper ~1300 m are displayed in Stoll et al. (2022). However, no particle size or chemical data has been published yet, hampering comparisons. This fact, and that SP data from ice cores is so scarce, make it very challenging to discuss this further as mentioned in the text.
Additionally, it is challenging to interrogate data obtained with other methods. At the current state, there is very limited knowledge of the number of nanoparticles within ice, and established methods focus either on the upper nanoscale or microscale. As such, we navigate within an uncharted territory. Furthermore, the number of particles, as well as data on mean sizes and composition, are massively biased by the lower detection limit. As such, our data and study should be regarded as a tentative approach to chart the nanoscale of particles found in polar ice cores. We will emphasise that we still have many blind spots, but aim to expand the accessible range of particles. As such, there are discrepancies, which, however, are negligible given the explorative approach suggested here.

Finally, a quick note to say I do not share Reviewer 1's concerns on contamination potential. Significant contamination from drill fluid or human handling would have shown up in the previous analyses. Drill fluid needs micro-cracks to penetrate into the ice core and these were not visualised. A clearer description of decontamination procedures and maybe a brief justification for their choice would be valuable. The second point highlighted by Reviewer 1, on the threshold setting, needs clarification before publication.

 A: We will improve the text on sample preparation and decontamination procedures. We will further explain the threshold procedures better as mentioned in our answer to the first referee report.

Minor suggestions:

The Introduction needs re-writing to streamline the information and argument presented. Many of the paragraphs reiterate arguments previously made.
A: We agree and will streamline the introduction to avoid repetitions.

L21: Is there not a more up-to-date reference than this 1997 one? The excellent Encyclopaedia of Quaternary Sciences by Koffman springs to mind (although I appreciate it is not OA).
A: Agreed, we will add more recent publications on the matter.

L32: please more be specific on the 'particular material characteristics of ice'!
A: This refers to the challenges provided by ice in solid form, such as the need for e.g. ice laboratories and specific cryo-sample holders as well as the logistical difficulties in obtaining and transporting samples. These challenges hamper the

straight-forward transfer of state-of-the-art analytical chemistry approaches to ice core samples. We will elaborate on this in the text.

L55: please explain what a "competitive trace element analysis" is.
A: This states that not only quadrupole MS, but also TOFMS can be used for trace elements analysis. We will edit this sentence.

L56-57: Please explain, for the average 'ice core' reader, the terms 'non-target particle screening' and 'the internal and external mixing state of particles' (if these advantages are actually relevant to this study).
A: Non-target screening enables rapid definition of a decision limit for all recorded m/z making it possible to choose specific elements with concentrations above a certain ppm-level for further analysis and without a priori knowledge. This reduces the processing time for TOF data by e.g. enabling to rapidly pinpoint relevant particulate elements beforehand. The exact workflow is described in Gonzalez de Vega et al (2023).
In atmospheric science, the **mixing state** of particles describes how different chemical components are distributed among individual aerosol particles. An **externally mixed** aerosol population consists of distinct particles, each composed of a single chemical species; thus, the particle population is chemically diverse at the single-particle level. In contrast, an **internally mixed** population contains particles that each comprise multiple chemical components, resulting in a more homogeneous chemical composition across the entire population. The mixing state significantly influences particle properties such as hygroscopicity, optical behaviour, and reactivity, and is therefore critical for understanding aerosol-climate and aerosol-health interactions. These states can vary a lot and are of importance in e.g. climate-relevant aerosol physical properties such as optical scattering/absorption and cloud condensation nuclei activity. This will be elaborated on in the revised manuscript.

Paragraph from L58: Isn't there some SP TOF work coming out of Ohio State (Stanislav Kutuzov)?
A: There is work being done at Ohio State, which has been presented at conferences over the last years. Unfortunately, it has not been published in peer-reviewed journals yet.

L103: Acid-cleaned vials?
A: No, we however used novel vials and checked blank levels to confirm the absence of contamination.

L125, 147: 24Mg etc are isotopes not elements. This occurs throughout the manuscript!
A: This is true and might appear slightly confusing. SPTOF analysis measures isotopic data, it is thus important to clarify which isotope is referred to. However, isotopic abundances are considered during calibration, which means that the Mg levels are determined over the 24Mg isotope. We will address this more clearly in the revised version.

L187: Again, isotopes are listed not elements. 43Ca in G7 is low on Figure 2 but 44Ca is not – surely the relative abundance of isotopes of the same element should be corrected for? Why not describe elements as elements? Why persist in using

isotopes? Hopefully Table 3 is not actually listing 56Fe/27Al (ditto for Figure 6)?
A: See answer above.

Figure 2: Last sentence of caption needs adjusting for clarity.
A: Changed.

L249: 23Mg should be 24 Mg.
A: Changed.

**References**

Gonzalez de Vega, R., E. Lockwood, T., Paton, L., Schlatt, L., and Clases, D.: Non-target analysis and characterisation of nanoparticles in spirits via single particle ICP-TOF-MS, *Journal of Analytical Atomic Spectrometry*, 38, 2656–2663, https://doi.org/10.1039/D3JA00253E, 2023.

Lockwood et al., (2021): An interactive Python-based data processing platform for single particle and single cell ICP-MS, *J. Anal. At. Spectrom.*, 2021, 36, 2536-2544, **https://doi.org/10.1039/D1JA00297J**

Lockwood et al., (2025): SPCal – an open source, easy-to-use processing platform for ICP-TOFMS-based single event data, *J. Anal. At. Spectrom.*, 2025, 40, 130-136, DOI https://doi.org/10.1039/D4JA00241E

Simonsen, M. F., Cremonesi, L., Baccolo, G., Bosch, S., Delmonte, B., Erhardt, T., Kjær, H. A., Potenza, M., Svensson, A., and Vallelonga, P.: Particle shape accounts for instrumental discrepancy in ice core dust size distributions, *Clim. Past*, 14, 601–608, https://doi.org/10.5194/cp-14-601-2018, 2018.

Stoll, N., Hörhold, M., Erhardt, T., Eichler, J., Jensen, C., and Weikusat, I.: Microstructure, micro-inclusions, and mineralogy along the EGRIP (East Greenland Ice Core Project) ice core – Part 2: Implications for palaeo-mineralogy, *The Cryosphere*, 16, 667–688, https://doi.org/10.5194/tc-16-667-2022, 2022.

Zeppenfeld et al., *Environ. Sci. Technol.* 2025, 59, 1, 328–336, https://doi.org/10.1021/acs.est.4c07098